# ECOASSISTANT: USING LLM ASSISTANT MORE AFFORDABLY AND ACCURATELY

## ABSTRACT

Today, users ask Large language models (LLMs) as assistants to answer queries that require external knowledge; they ask about the weather in a specific city, about stock prices, and even about where specific locations are within their neighborhood. These queries require the LLM to produce code that invokes external APIs to answer the user's question, yet LLMs rarely produce correct code on the first try, requiring iterative code refinement upon execution results. In addition, using LLM assistants to support high query volumes can be expensive. In this work, we contribute a framework, EcoAssistant, that enables LLMs to answer code-driven queries more affordably and accurately. EcoAssistant contains three components. First, it allows the LLM assistants to converse with an automatic code executor to iteratively refine code or to produce answers based on the execution results. Second, we use a hierarchy of LLM assistants, which attempts to answer the query with weaker, cheaper LLMs before backing off to stronger, expensive ones. Third, we retrieve solutions from past successful queries as in-context demonstrations to help subsequent queries. Empirically, we show that EcoAssistant offers distinct advantages for affordability and accuracy, surpassing GPT-4 by 10 points of success rate with less than $50\%$ of GPT-4's cost.

## 1 INTRODUCTION

Recently, users have been using conversational LLMs such as ChatGPT (OpenAI, 2023) for various queries. Reports indicate that $23\%$ of ChatGPT user queries are for knowledge extraction purposes (Fishkin, 2023). Many of these queries require knowledge that is external to the information stored within any pre-trained large language models (LLMs). For example, users ask about the weather in their city: "What is the current cloud coverage in Mumbai, India?"; they ask about stock prices: "Can you give me the opening price of Microsoft for the month of January 2023?"; some even ask for place recommendations: "I'm looking for a 24-hour pharmacy in Montreal, can you find one for me?". These tasks can only be completed by calling external APIs that contain the requested information. As such, these types of tasks—what we call *code-driven question answering*—require LLMs to generate code to fetch necessary information via APIs.

Just as human coders rarely generate correct code on the first attempt, LLMs also struggle (Yang et al., 2023). This is especially dire since current LLMs lack the ability to execute their generated code and iteratively debug as most human programmers do. In addition, as ChatGPT received roughly $80M$ queries in July 2023, $23\%$ of it makes up $4M$ knowledge queries that month itself (Fishkin, 2023; Chen et al., 2023; Wang et al., 2023a). Such a high volume of user queries can be expensive for those who aim to develop a system using online LLM services with a fee to process these queries.

To overcome these challenges, in this work, we present EcoAssistant, the first system that is tailored for leveraging conversational LLMs to tackle code-driven question answering more affordably and accurately. EcoAssistant doesn't need any offline preparation or training; it is a purely online service that improves with use. It contains three fundamental components. First, to support iterative coding, it allows the conversational LLM as an assistant agent to converse with an automatic code executor and iteratively refine code to make the correct API calls. We build EcoAssistant using AutoGen (Wu et al., 2023), a recent framework that enables building LLM applications via multi-agent conversation. Unlike existing practices that use LLMs to produce code or answer the

user query in a single generation, our system design exploits the recent advance of conversational LLMs that can iteratively refine their outputs (OpenAI, 2023; Touvron et al., 2023).

Second, we employ a hierarchy of LLM assistants, referred to as assistant hierarchy, which attempts to answer the query with weaker, cheaper LLMs before backing off to stronger, expensive ones. For each query, we start the conversation with the most cost-effective LLM assistant, and progressively back off to more expensive ones only when the current one fails. As LLMs typically have heterogeneous pricing structures, such a simple strategy could reduce the overall cost of the system by reducing the usage of expensive LLMs.

Third, we propose solution demonstration, which retrieves solutions from past successful queries as in-context demonstrations to help subsequent queries. To achieve this, we store correct query-code pairs in a database once a query succeeds; then when a new query enters, we retrieve the most similar query as well as the associated code from the database as in-context demonstrations in the LLM's prompt. With the proven solutions demonstrated, the assistant is more likely to generate accurate and efficient responses without redundant iterations, thereby increasing the likelihood of success.

Although the assistant hierarchy and solution demonstration offer distinct advantages when used individually, we find that their interplay leads to a synergistic effect that amplifies their individual benefits. This is, because the assistants in the hierarchy share the database storing query-code pairs, these solutions from stronger, expensive LLMs serve as useful guidance for the weaker models on subsequent queries. As a consequence, the weaker assistant is likely to solve more queries in the future, which further reduces the systems' reliance on expensive LLMs.

We conduct systematic experiments on various types of queries to investigate both the performance and the dollar cost of the proposed system. Our results highlight that the assistant hierarchy can significantly reduce the cost, while the solution demonstration largely boosts the system's performance. In addition, `EcoAssistant`, which incorporates both of these strategies, achieves superior performance with a further reduction of the cost. In addition, we show that `EcoAssistant` outperforms an individual GPT-4 assistant with a margin of 10% success rate with less than half of the expense.

## 2 THE TASK OF CODE-DRIVEN QUESTION ANSWERING

In this work, we focus on a practical yet challenging task called *code-driven question answering*, where LLMs have to answer knowledge queries; The LLM has to generate code to invoke APIs for acquiring the necessary information needed to answer the user's question. For example, a user query could be asking for dynamic or real-time information like the weather of a specific location at a certain date. Since this information is not stored in the model's internal knowledge or general knowledge base, the model would rely on the weather APIs to acquire the information. To achieve this, LLMs need to not only understand the user's query correctly but also write decent Python code. Thus, this new task presents a multi-faceted challenge: it demands proficiency in language understanding and generation of both natural and programming language. This characteristic differentiates code-driven question answering from existing question answering paradigms such as open-domain question answering (Lee et al., 2019; Chen et al., 2017) or browser-assistant question answering (Nakano et al., 2021), since they typically do not challenge the LLMs' capability of generating and refining code. It is also different from generic code generation task (Chen et al., 2021; Hendrycks et al., 2021; Austin et al., 2021; Lu et al., 2021; Yang et al., 2023) by requiring LLMs to exploit domain-specific API based on the user query.

**Iterative coding.** Code-driven question answering naturally requires iterative coding (Yang et al., 2023). We connect the underlying LLM attempting to generate the code with a code executor. Intuitively, the code executor executes the generated code and forwards either the execution results or the failed execution trace back to the LLM. This interaction may occur multiple times, as the LLM uses the previous execution trace to refine its generation. One could view this process as an automatic multi-turn chat between the LLM and the code executor, which happens completely in the background, without the user's involvement. We adopt chat LLMs such as GPT-3.5-turbo, allowing us to leverage all the recent advancements of LLMs for chat-purposes.

**Queries come streaming.** We also consider a real-world scenario where queries come streaming sequentially over time. Therefore, each query is not an independent task but could leverage past

queries as guidance. In such a setting, one could imagine deriving keeping track of successful queries to improve future ones. Our system, described below, investigates how to utilize past queries to better serve future ones.

# 3 ECOASSISTANT: USING LLM ASSISTANT MORE AFFORDABLY AND ACCURATELY

To re-iterate, the task of code-driven question answering is both challenging and expensive. LLMs struggle to generate the correct code at the first attempt to utilize APIs, and handling a high volume of user queries using LLM services with a fee can be cost-intensive. To tackle this task in an affordable and accurate manner, we develop `EcoAssistant`, a system that uses LLMs to answer knowledge queries correctly while reducing dollar costs.

`EcoAssistant` contains three components (see Figure 1). First, it places LLMs as an assistant agent in conversation with a code executor. The LLM iteratively debugs its code by reading the code executor's outputs or failed execution trace, and finally produces the answer based on the information obtained. Second, to reduce expenses, we use a hierarchy of LLM assistants, attempting queries with cheaper LLM assistants before resorting to more expensive alternatives. Third, we keep track of successful queries and the associated code and use them as in-context demonstrations for subsequent ones. This allows LLMs in the future to use past successes as guidance. Our system requires no offline preparation, no dataset curation, and no training.

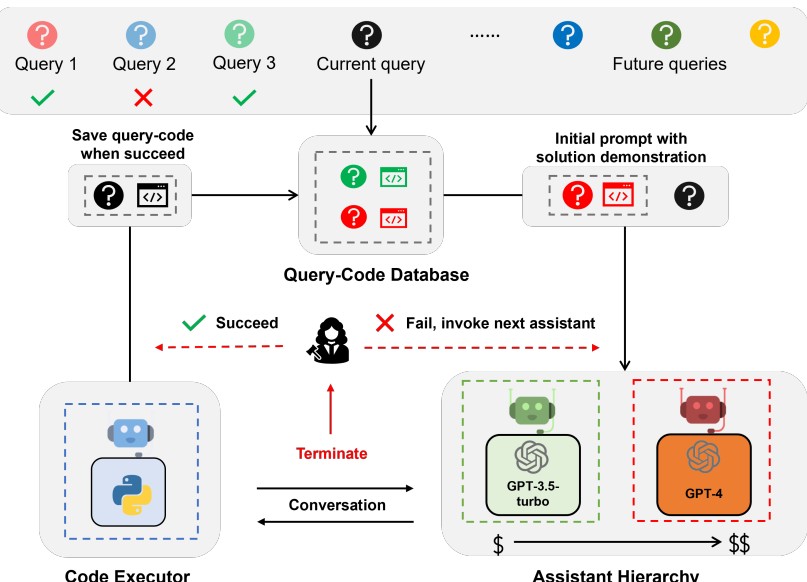

Figure 1: `EcoAssistant`: the system involves two agents, one executor agent for executing the code and the other assistant agent backed by LLMs for suggesting code to obtain information and address the user queries. The query-code database stores the previous successful query and code pair. When a new query comes, the most similar query in the database is retrieved and then demonstrated in the initial prompt with the associated code. The conversation invokes the most cost-effective assistant first and tries the more expensive one in the assistant hierarchy only when the current one fails.

**Automated conversation between LLM assistants and code executor.**   `EcoAssistant` places the LLM as an assistant agent within a conversation with a code executor. The executor extracts the generated code and executes it, forwarding the output back to the LLM; it then awaits the next conversation turn, where presumably the LLM will refine its generation, learning from its past mistake, or produce the final answer according to the execution results.

To achieve this conversation flow, we develop our system upon AutoGen (Wu et al., 2023), a recent infrastructure that facilitates automated multi-agent conversation. In particular, we leverage the

built-in `AssistantAgent` and `UserProxyAgent` of AutoGen as the LLM assistant and code executor, respectively. The former is configured with the dedicated system prompts proposed in AutoGen, which instructs the LLM to 1) suggest code in a coding block when necessary, 2) refine the code according to the execution results, and 3) append a special code "TERMINATE" at the end of the response when it wants to terminate the conversation.

The latter, acting as a proxy of the user, automatically extracts the code from the LLM's message, and executes it in the local environment. It then sends the execution results back to the LLM. When there is no code detected, it would send back a default message. Thus, the conversation is automated and the user only needs to input the original query to trigger the conversation without manual intervention like copying, pasting, and executing the code.

Finally, the conversation terminates when encountering one of the following cases: 1) the context window of the LLM is exceeded; 2) the number of back-forth turns in the conversation exceeds a set threshold[1]; and 3) the LLM appends "TERMINATE" at the end of its response.

**Assistant hierarchy.** We employ a hierarchy of LLM assistants. In particular, given multiple LLMs, we initiate an `AssistantAgent` for each and start the automated conversation with the most cost-effective LLM assistant. If the conversation between the current LLM assistant and the code executor concludes without successfully resolving the query, the system would then restarts the conversation with the next more expensive LLM assistant in the hierarchy. Considering that LLMs typically have varied pricing structures (*e.g.*, GPT-3.5-turbo is an order of magnitude cheaper than GPT-4), this strategy has the potential to significantly reduce costs by minimizing the usage of expensive LLMs, while still effectively addressing queries.

**Solution demonstration.** In most practical scenarios, queries from users would appear sequentially over time. Our system leverages past success to help the LLM assistants address future queries. Specifically, whenever a query is deemed successfully resolved by user feedback, we capture and store the query and the final generated code snippet. These query-code pairs are saved in a specialized vector database. When new queries appear, `EcoAssistant` retrieves the most similar query from the database, which is then appended with the associated code to the initial prompt for the new query, serving as a demonstration. We show that this utilization of past successful query-code pairs improves the query resolution process with fewer iterations and enhances the system's performance.

The assistant hierarchy and solution demonstration as standalone designs offer distinct advantages: the assistant hierarchy has the potential to reduce the cost by minimizing the reliance on expensive LLMs, and the solution demonstration can enhance the performance of LLMs by leveraging past success. Together, they amplify the individual benefits. Even without a specialized design, the stronger LLM assistants implicitly advise weaker ones in the hierarchy by sharing their solutions via the query-code database.

## 4 EXPERIMENT

In this section, we conduct experiments to investigate the performance and dollar cost of the `EcoAssistant` on various types of queries, with both model evaluation and human evaluation. We empirically show the individual benefits introduced by the assistant hierarchy and the solution demonstration, and the `EcoAssistant` could surpass an individual GPT-4 assistant by 10% of the success rate with less than half of the GPT-4 assistant's expense.

### 4.1 SETUP

**Dataset** We consider three datasets from the ToolBench (Qin et al., 2023) whose queries correspond to the domains of **Places**, **Weather**, and **Stock** respectively. We randomly sample 100 queries from each dataset. Each dataset comes with a recommended API to use. We list an example query and the API for each dataset in Table 1. In addition to evaluating the methods on each dataset separately, we also consider a setup where all three datasets are combined, resulting in 300 queries. We then randomly shuffle the 300 queries to construct three datasets with different orders of queries and refer to them as **Mixed-1**, **Mixed-2**, and **Mixed-3** respectively.

---

[1] We set the maximum number of turns of a conversation to 5 in this paper.

Table 1: The default API and example query for each dataset.

| Dataset | API | Example query |
|---------|-----|---------------|
| Places | Google Places[1] | *I'm looking for a 24-hour pharmacy in Montreal, can you find one for me?* |
| Weather | Weather API[2] | *What is the current cloud coverage in Mumbai, India?* |
| Stock | Alpha Vantage Stock API[3] | *Can you give me the opening price of Microsoft for the month of January 2023?* |

[1] `https://developers.google.com/maps/documentation/places/web-service/overview`
[2] `https://www.weatherapi.com`
[3] `https://www.alphavantage.co/documentation/`

**Prompt and LLMs**  The initial prompt contains the query, the API name/key, and the retrieved query-code pair when solution demonstration is used, and LLMs have to rely on their internal knowledge of the API and coding skills to answer the query. To avoid leaking the confidential API key to the LLM services, we randomly generate a unique four-byte text string in hexadecimal for each API as a fake key, then whenever the code is executed, we replace the fake key in the code with the actual API key so that the code can function as expected. We present the prompt we used in the experiments in Appendix D. Since the techniques we propose are orthogonal to existing prompting methods and can be synergistically applied, we also conducted experiments using the Chain-of-Thought (CoT) prompting (Wei et al., 2022). In this work, we focus on conversational LLMs including two black-box models with various costs (GPT-3.5-turbo and GPT-4) and one open-source model (LLAMA-2-13B-chat (Touvron et al., 2023)). We assume the dollar cost of the LLAMA-2-13B-chat is zero since it can be hosted with a reasonable amount of computing resources, while recording the cost of using black-box LLM services.

**Compared methods and implementation**  We investigate the performance of three assistants backed by different conversational LLMs (LLAMA-2-13B-chat, GPT-3.5-turbo, and GPT-4) as well as two types of assistant hierarchy: AssistantHier-G (GPT-3.5-turbo + GPT-4) and AssistantHier-L (LLAMA-2-13B-chat + GPT-3.5-turbo + GPT-4). For each assistant or assistant hierarchy, we include its vanilla version and the following variants: + CoT (with Chain-of-Thought prompting), + SolDemo (with solution demonstration), and + CoT + SolDemo (with both Chain-of-thought prompting and solution demonstration). We couple each assistant or assistant hierarchy to a code executor agent in order to tackle the code-driven question answering task. Note that the proposed `EcoAssistant` system includes both assistant hierarchy and solution demonstration *i.e.*, AssistantHier-G/L (+ CoT) + SolDemo. We implement all the systems based on AutoGen (Wu et al., 2023), a Python library[2] for multi-agent conversation framework. For solution demonstration, we use Chroma (Chroma, 2023), an open-source embedding database to store the query-code pairs; we use multi-qa-mpnet-base-dot-v1 model to embed the user query and the cosine similarity for similarity search [3].

**Evaluation protocol**  We focus on both the dollar cost and performance of compared methods. For model performance, we report the success rate, *i.e.*, the percentage of queries that are successfully handled by the model. Since the queries usually do not have ground truth answer (Qin et al., 2023), *e.g.*, asking the weather at a certain date, we adopt both model evaluation and human evaluation. For model evaluation, we leverage GPT-4 as a proxy of the user to judge whether the system successfully handles the query (Zheng et al., 2023; Fu et al., 2023; Wang et al., 2023b). In particular, after the conversation is terminated, we prompt GPT-4 with the whole conversation history and ask it whether the tested system successfully handles the query. The details can be found in Appendix D. We repeat each evaluation three times with different random seeds if not otherwise specified.

## 4.2 MODEL EVALUATION: INDIVIDUAL DATASET

First, we conduct experiments on each of the three individual dataset to investigate the performance of compared systems. The results are present in Table 2. We summarize our findings as below.

**Finding 1: the compared LLMs have distinct performance- the more expensive the model is, the better it performs.**  From the results, we can see that individual LLMs (GPT-3.5-turbo, GPT-4, LLAMA-2-13B-chat) have heterogeneous performance. In particular, GPT-4 achieves the

---

[2] `https://github.com/microsoft/autogen`
[3] `https://huggingface.co/sentence-transformers/multi-qa-mpnet-base-dot-v1`

Table 2: Success rate (%) and dollar cost on the Places, Weather, and Stock dataset.

| Method | Places | | Weather | | Stock | |
|---|---|---|---|---|---|---|
| | Success rate (%) | Cost | Success rate (%) | Cost | Success rate (%) | Cost |
| LLAMA-2-13B-chat | 27.00 | 0.00 | 6.33 | 0.00 | 6.67 | 0.00 |
| + CoT | 25.00 | 0.00 | 7.67 | 0.00 | 6.33 | 0.00 |
| + SolDemo | 56.00 | 0.00 | 6.00 | 0.00 | 31.33 | 0.00 |
| + CoT + SolDemo | 52.00 | 0.00 | 4.67 | 0.00 | 14.00 | 0.00 |
| GPT-3.5-turbo | 39.33 | 0.47 | 46.00 | 0.41 | 17.00 | 0.36 |
| + CoT | 61.33 | 0.67 | 69.00 | 0.67 | 50.00 | 0.84 |
| + SolDemo | 77.33 | 0.49 | 79.67 | 0.54 | 68.00 | 0.50 |
| + CoT + SolDemo | 70.33 | 0.73 | 78.33 | 0.69 | 64.67 | 0.80 |
| GPT-4 | 85.00 | 12.58 | 87.33 | 10.73 | 59.33 | 18.49 |
| + CoT | 78.67 | 15.16 | 75.67 | 11.67 | 57.67 | 19.01 |
| + SolDemo | 88.00 | 11.76 | 87.33 | 10.81 | 75.00 | 14.33 |
| + CoT + SolDemo | 87.33 | 13.75 | 84.67 | 11.30 | 77.67 | 15.52 |
| AsistantHier-G | 89.33 | 8.61 | 90.67 | 6.21 | 64.33 | 15.99 |
| + CoT | 89.33 | 7.28 | 88.33 | 4.87 | 73.33 | 11.85 |
| + SolDemo | 96.67 | 3.73 | 95.00 | 3.04 | 81.67 | 8.10 |
| + CoT + SolDemo | 96.33 | 5.52 | 93.00 | 3.49 | 86.00 | 8.04 |
| AsistantHier-L | 91.67 | 5.97 | 91.67 | 5.89 | 66.33 | 15.10 |
| + CoT | 91.33 | 5.89 | 89.00 | 4.36 | 75.33 | 11.01 |
| + SolDemo | 97.00 | 3.33 | 98.00 | 2.24 | 85.00 | 6.70 |
| + CoT + SolDemo | 95.33 | 3.52 | 96.33 | 2.82 | 84.33 | 6.86 |

highest success rate followed by GPT-3.5-turbo, while LLAMA-2-13B-chat underperforms the others. On the other hand, GPT-4 has the highest cost, GPT-3.5-turbo is relatively cost-effective, and LLAMA-2-13B-chat has zero dollar cost since it can be hosted on a local machine.

**Finding 2: the Chain-of-Thought (CoT) prompting could largely enhance LLM with moderate performance, *i.e.*, GPT-3.5-turbo.**   the Chain-of-Thought (CoT) prompting consistently enhances the performance of GPT-3.5-turbo across datasets. However, for GPT-4 and LLAMA-2-13B-chat, the success rate doesn't necessarily benefit from CoT. We hypothesize this could be because GPT-4 is already highly competent, leaving little room for CoT to further improve its performance, while the performance of LLAMA-2-13B-chat does not benefit from CoT probably due to its inherent inadequacy in tackling code-driven question answering tasks. In addition, we find that CoT tends to increase the dollar cost, since it encourages the model to think step by step and therefore would increase the number of tokens that LLMs input and output. Finally, when comparing AssistantHier-G to AssistantHier-G + CoT, the latter results in a reduced cost. This is attributed to CoT enhancing the success rate of GPT-3.5-turbo, therefore decreasing the reliance on the more expensive GPT-4.

**Finding 3: solution demonstration could boost the success rate, especially when the method is not well-performing.**   In almost all cases, solution demonstration could significantly improve the success rate without introducing a notable increase in the cost, especially for less competitive LLMs. In particular, solution demonstration almost doubles the success rate of vanilla GPT-3.5-turbo on the Places and the Weather dataset, while introducing a 3x boost of success rate on the Stock dataset. For LLAMA-2-13B-chat, solution demonstration also increases the success rate on the Places and the Stock dataset by a large margin. Yet for high-performing model GPT-4, solution demonstration does not result in a significant performance boost except for the Stock dataset where the GPT-4 only exhibits a success rate of around 60%.

**Finding 4: Compared to the GPT-4, the assistant hierarchy could significantly reduce the cost while slightly improving the success rate.**   By comparing assistant hierarchy (AsistantHier-G and AsistantHier-L) with vanilla GPT-4, we can see that the cost is significantly reduced; in particular, AsistantHier-G achieves cost savings of approximately 10%-30%, while AsistantHier-L realizes reductions in the range of 15%-50%. In addition, assistant hierarchy also leads to a slight boost in success rate since it allows trying multiple assistants for a single query.

**Finding 5: `EcoAssistant` (assistant hierarchy + solution demonstration) achieves superior performance with moderate cost.**   Finally, `EcoAssistant` (AsistantHier-G/L (+ CoT) + Sol-Demo) delivers the highest success rate across all datasets, surpassing the top-performing GPT-4 variants by roughly a 10 percentage point margin in success rate. Additionally, this combination leads to a further cost reduction of approximately 30%-50% when compared to solely using the

assistant hierarchy approach (AsistantHier-G/L (+ CoT)). These findings are surprising, given that the solution demonstration could only enhance the success rate when used individually. We attribute this synergy to the fact that solutions generated by the high-performing GPT-4 subsequently guide the more affordable, albeit weaker, LLMs. As a result, cost savings emerge, because these more economical LLMs can tackle a greater number of tasks using the GPT-4 solution as a demonstration, thus minimizing the reliance on the pricier GPT-4 assistant.

## 4.3 MODEL EVALUATION: MIXED DATASET

We also evaluate methods on mixed datasets: **Mixed-1**, **Mixed-2**, and **Mixed-3**. Each of them encompasses queries from all individual datasets, distinguished by different query orderings. These experiments investigate how the methods perform when queries span multiple domains and ordering. Specifically, we assess six methods in this experiment: GPT-3.5-turbo, GPT-3.5-turbo + SolDemo, GPT-4, GPT-4 + SolDemo, AsistantHier-G, AsistantHier-G + SolDemo (`EcoAssistant`). We visualize the results in Figure 2 in order to demonstrate the scaling trend of cost and number of successful queries with regard to the number of queries processed in the streaming setting.

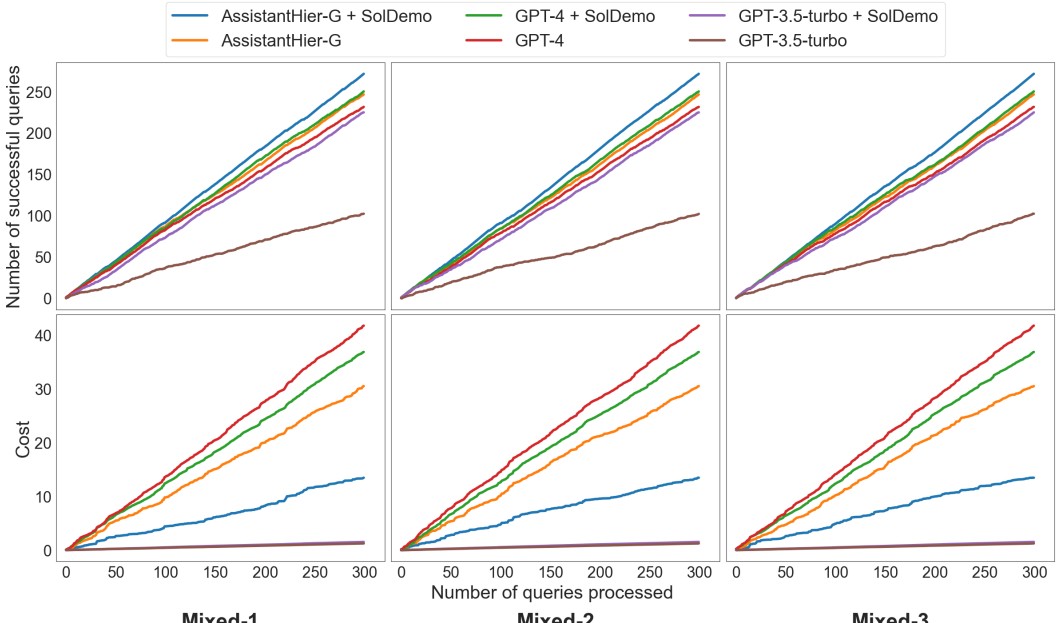

Figure 2: The curves of the number of successful queries / cost with regard to the number of queries processed on mixed datasets. The three datasets encompass queries from all individual datasets, distinguished by different query orderings. We can see that `EcoAssistant` (AsistantHier-G + SolDemo) leads to the best performance while maintaining relatively low cost.

From the results, we draw the following conclusions. First, because the solution demonstration relies on the past solutions and therefore could be affected by the ordering of the queries, it is then important to ascertain its robustness against different query sequences; the results suggest that the method is largely order-agnostic, as evidenced by the consistency in performance curves (X + SolDemo) across datasets. Second, the two variants of GPT-3.5-turbo are the most cost-effective since the cost curves are nearly flat compared to other methods, yet the GPT-3.5-turbo alone underperforms all the compared methods in terms of the success count; however, integrating solution demonstration (GPT-3.5-turbo + SolDemo) substantially uplifts its efficacy. Third, despite the descent performance GPT-4 renders, it is the most cost-intensive method as indicated by its steeper cost curve; fortunately, the assistant hierarchy (AsistantHier-G) would reduce the cost as its cost curves have a smaller slop, without sacrificing the performance. Finally, the `EcoAssistant` (AsistantHier-G + SolDemo) exhibits the best performance and concurrently has a much flatter cost curve than other methods, except for GPT-3.5-turbo variants.

### 4.4 HUMAN EVALUATION: MIXED DATASET

For human evaluation, we sample 100 queries from all the 300 queries of different datasets to form a dataset referred to as **Mixed-100**. We gather one code snippet for each of the 100 queries, which could produce the necessary information for addressing the query. In particular, for each query, we collect all the code snippets generated by LLMs in previous experiments. From this collection, we manually choose and validate one snippet (with necessary manual modifications) to ensure that the code snippet can effectively obtain the needed information. Thus, one can refer to the output of the golden code snippet when assessing whether the model successfully addresses the query. For all the experiments in this section, we adopt this strategy to do human evaluation.

Table 3: Human evaluation results on the **Mixed-100** dataset.

| | GPT-3.5-turbo | GPT-3.5-turbo + SolDemo | GPT-4 | GPT-4 + SolDemo | AsistantHier-G | AssistantHier-G + SolDemo |
|---|---|---|---|---|---|---|
| **Metric** | **Main results** | | | | | |
| Success rate (%) | 25 | 45 | 59 | 78 | 63 | 80 |
| Cost | 0.36 | 0.48 | 13.77 | 10.27 | 11.84 | 5.90 |
| **Model** | **Avg. model calls per query** | | | | | |
| GPT-3.5-turbo | 2.42 | 2.91 | - | - | 2.42 | 2.92 |
| GPT-4 | - | - | 3.12 | 2.57 | 2.51 | 1.25 |

From the main results part of Table 3, we can see that our main conclusion still holds for the case of human evaluation. Specifically, solution demonstration can significantly improve the success rate, and assistant hierarchy contributes to cost savings when compared to GPT-4. More importantly, `EcoAssistant` (AssistantHier-G + SolDemo) delivers top-tier performance at a moderate expense; when comparing it against GPT-4, we can see that its success rate is 10 points higher while incurring less than half of GPT-4's cost. To better explain the effect of the proposed techniques, we also present the averaged model calls per query for each method in Table 3. First, the solution demonstration increases the model calls of GPT-3.5-turbo, because the vanilla GPT-3.5-turbo struggles to produce formatted code that can be extracted and executed by the code executor, and therefore the conversation terminates early without the query being appropriately addressed; with solution demonstration, GPT-3.5-turbo is more likely to generate formatted code for the code executor to execute, thus the conversation would proceed. Second, for GPT-4, the solution demonstration reduces the number of model calls because it guides the model to write good code at the beginning, requiring fewer turns to refine the code for outputting necessary information. Finally, comparing AssistantHier-G with/without SolDemo, we can see that the averaged model calls of GPT-3.5-turbo increase while that of GPT-4 reduces. This indicates that `EcoAssistant` (AssistantHier-G + SolDemo), even with a higher success rate, relies less on expensive GPT-4 because the GPT-3.5-turbo is able to address more queries thanks to the solution demonstration, leading to the saving of cost.

### 4.5 HUMAN EVALUATION: AUTONOMOUS SYSTEMS WITHOUT HUMAN FEEDBACK

In the above experiments, we incorporate a user in the loop—either an actual human or a GPT-4 model—to determine the successful completion of a query. This feedback serves three primary functions: 1) calculating the success rate of the evaluated method; 2) for solution demonstration to decide whether to store the query-code pair; and 3) for assistant hierarchy to decide whether to invoke the next assistant in the hierarchy. However, in practice, users may prefer a system that operates autonomously, without necessitating user feedback. Regarding this, we build an autonomous system for each compared method, which requires no human feedback. Specifically, we add a GPT-4 evaluator to serve the aforementioned functions 2) and 3), and after all the queries are processed, we manually assess the success of each query to calculate the success rate. In addition, because now we treat the GPT-4 evaluator as part of the system, we include its cost as the system cost as well. Note that the method without solution demonstration or assistant hierarchy (*e.g.*, GPT-3.5-turbo alone) would remain the same as before.

We evaluate these autonomous systems on the **Mixed-100** dataset, applying the same human evaluation strategy as in the previous section. The outcomes are detailed in Table 4. When comparing `EcoAssistant` (AssistantHier-G + SolDemo) to GPT-4, we still observe a success rate boost exceeding 10 points and a cost reduction of over 50%. However, while `EcoAssistant` presents a cost comparable to its non-autonomous counterpart (as shown in Table 3), its success rate is dimin-

ished by 8 points. This is because the GPT-4 evaluator would occasionally mistrust the GPT-3.5-turbo and would not resort to the GPT-4 assistant, leading to the performance drop; simultaneously, as it calls upon the GPT-4 assistant less frequently than its non-autonomous version, the cost does not grow even when factoring in the GPT-4 evaluator's expenses.

We also present the run-time of these autonomous systems in Table 4. We can see that the solution demonstration can largely reduce the run-time, indicating that it streamlines the query resolution process with fewer iterations. In addition, the assistant hierarchy (AssistantHier-G) exhibits the longest run-time, as it often tries out both GPT-3.5-turbo and GPT-4 assistant. Remarkably, `EcoAssistant` necessitates less than half of the run-time of AssistantHier-G and even outperforms a standalone GPT-4 assistant. This underscores the synergistic effect of integrating the assistant hierarchy and solution demonstration, further reducing the dependence on the more latency-prone GPT-4 assistant.

Table 4: Human evaluation results on the **Mixed-100** for autonomous systems.

| | GPT-3.5-turbo | GPT-3.5-turbo + SolDemo | GPT-4 | GPT-4 + SolDemo | AssistantHier-G | AsistantHier-G + SolDemo |
|---|---|---|---|---|---|---|
| **Metric** | **Main results** | | | | | |
| Success rate (%) | 25 | 47 | 59 | 77 | 54 | 72 |
| Cost | 0.36 | 2.46 | 13.77 | 12.07 | 12.99 | 5.78 |
| **Model** | **Avg. model calls per query** | | | | | |
| GPT-3.5-turbo | 2.42 | 2.84 | - | - | 2.45 | 2.90 |
| GPT-4 | - | - | 3.12 | 2.49 | 2.29 | 0.59 |
| Run-time (s) | 2414 | 2073 | 5272 | 3873 | 8993 | 4033 |

## 5 RELATED WORK

Here, we briefly discuss related work of existing attempts that build LLM systems/applications upon multi-agent conversation and prior works on cost-effective deployment of LLMs.

**LLM-based multi-agent conversation**   LLM-based agents have attracted great attention from both practitioners and researchers (Xi et al., 2023; Wang et al., 2023c; Liu et al., 2023). Recently, there have been efforts towards harnessing multi-agent conversations in LLM-based applications to unleash the potential of among-agent collaboration (Wu et al., 2023). Example applications include collaborative task completion with multiple agents (Li et al., 2023; Hong et al., 2023; Qian et al., 2023; Talebirad & Nadiri, 2023) and leveraging multi-agent debate to encourage divergent thinking (Liang et al., 2023) or to improve factuality and reasoning (Du et al., 2023). In this work, we focus on exploiting the multi-agent conversation framework to tackle code-driven question answering with an emphasis on both cost-efficiency and performance.

**Cost-effective deployment of LLMs**   Countless efforts have been devoted to the cost-effective deployment of LLMs. Most of the existing attempts aim to improve the time/compute efficiency via techniques like model quantization (Yao et al., 2023) and prompt summarization (Arefeen et al., 2023), compression (Mu et al., 2023), and batching (Lin et al., 2023), *etc.*. In contrast, we seek to reduce the dollar cost of using LLM API services. With a similar goal, EcoOptiGen (Wang et al., 2023a) strives to reduce dollar cost for hyperparameter optimization of LLM inference and FrugalGPT (Chen et al., 2023) explores several techniques to reduce the dollar cost for single-turn text generation, while we focus on leveraging LLM as agent in a multi-agent conversation in a cost-effective way.

## 6 CONCLUSION

In this study, we explore affordable and precise LLM applications for code-driven question answering. We introduce `EcoAssistant`, an LLM-based system built upon a two-agent conversation framework. It involves an assistant agent backed by LLM and a code executor agent, and relies on their interaction to address user queries. `EcoAssistant` also includes two simple yet effective techniques: assistant hierarchy, which prioritizes cost-effective LLMs, and solution demonstration, which leverages past successful solutions for new queries. Our empirical evaluations demonstrate that `EcoAssistant` could simultaneously reduce the cost and enhance the performance.

ETHICS STATEMENT

The proposed systems would execute the code suggested by LLMs, which could be risky. Users should carefully consider the potential risks and ensure that the system runs in an isolated environment. In addition, letting LLMs write code to call specific APIs may cause a leak of the API key. To avoid this, our system replaces the actual API key with a random token to keep the API keys confidential and not visible to LLMs.

REPRODUCIBILITY STATEMENT

An anonymized repository containing the source code and data is provided with instructions for reproducing the experiments reported in this paper: `https://anonymous.4open.science/r/EcoAssistant-submission`.

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

## A   LIMITATIONS AND FUTURE WORK

**Limitations**   First, our system relies on a pre-defined hierarchy of LLM assistants. Such a static hierarchy may not always be optimal for all queries, and the system may benefit from a more adaptive selection mechanism. Second, our system depends on a database to store past successful query-code pairs, it may become a bottleneck when processing millions of queries. Thus, some pruning mechanisms that delete less useful items in the database could be helpful when the number of queries explodes. Third, while `EcoAssistant` attempts to handle a broad range of queries, it might not be as adept at deeply specialized or niche queries that demand expert-level domain knowledge. Fourth, while `EcoAssistant` attempts to handle a broad range of queries, it might not be as adept at deeply specialized or niche queries that demand expert-level domain knowledge. In addition, the back-and-forth conversational nature, especially with multiple iterative refinements, might introduce latency, leading to longer response times for the end users. Finally, the system might struggle with long conversational contexts, especially given the token limits of current LLMs. This could affect the quality of responses in extended conversations.

**Future work**   Here, we list several directions for future work. 1) Informative user feedback: In this work, we leverage binary user feedback indicating whether the user query is successfully addressed. Incorporating more informative user feedback to guide the conversation in `EcoAssistant` might enable more targeted and efficient task completion. 2) More agents in the system: In this work, we involve two types of agents in the system. One direction to explore could be adding more agents to the system for better collaborative task completion. 3) Advanced retrieval mechanisms: The current approach retrieves past solutions based on query similarity. Exploring more advanced retrieval mechanisms might enhance the efficacy of solution demonstrations. 4) Multimodal interactions: Extending `EcoAssistant` to support multimodal interactions, such as voice or images, can broaden its applicability and cater to an even wider user base.

## B   MODEL EVALUATION V.S. HUMAN EVALUATION

Here, we investigate the reliability of model (GPT-4) evaluation. In particular, we treat whether the query is successfully addressed as a binary classification task, and use the human evaluation results as ground truth to assess the efficacy of model evaluation results. We evaluate the accuracy, precision, and recall of model evaluation using queries and the human evaluation results in Section 4.4. The results can be found in Table 5; we separate the results based on the methods used to process the queries. From the results, we can see that all the recall is 100%, which means when the GPT-4 evaluator concludes that the assistant fails, it truly fails, while the precision ranges from 66% to 84%, indicating that there is still room for improvement. As the autonomous systems described in Section 4.5 rely on the GPT-4 evaluator to judge whether the user query is successfully addressed, a better GPT-4 evaluator is likely to contribute to better autonomous systems, which we leave as future work.

Table 5: We evaluate the efficacy of model evaluation using the human evaluation results as ground truth for experiments in Section 4.4. We found that model evaluation is always correct when it concludes that the assistant fails to address the user query, as all the recall is 100%, while the precision ranges from 66% to 84%, which indicates that model evaluation could provide a certain level of signal on the system performance.

| Metric | GPT-3.5-turbo | GPT-3.5-turbo + SolDemo | GPT-4 | GPT-4 + SolDemo | AsistantHier-G | AsistantHier-G + SolDemo |
|---|---|---|---|---|---|---|
| Accuracy | 91.75 | 82.63 | 80.41 | 85.86 | 85.88 | 71.72 |
| Precision | 75.76 | 72.58 | 75.32 | 84.78 | 72.09 | 66.12 |
| Recall | 100 | 100 | 100 | 100 | 100 | 100 |

## C   A VISUALIZATION OF THE ASSISTANT-CODE EXECUTOR CONVERSATION

Here, we demonstrate the assistant-code executor conversation in Figure 3. In particular, we trigger the conversation with a user query. Then the conversation between the LLM assistant and the code executor would proceed automatically until a termination condition is satisfied.

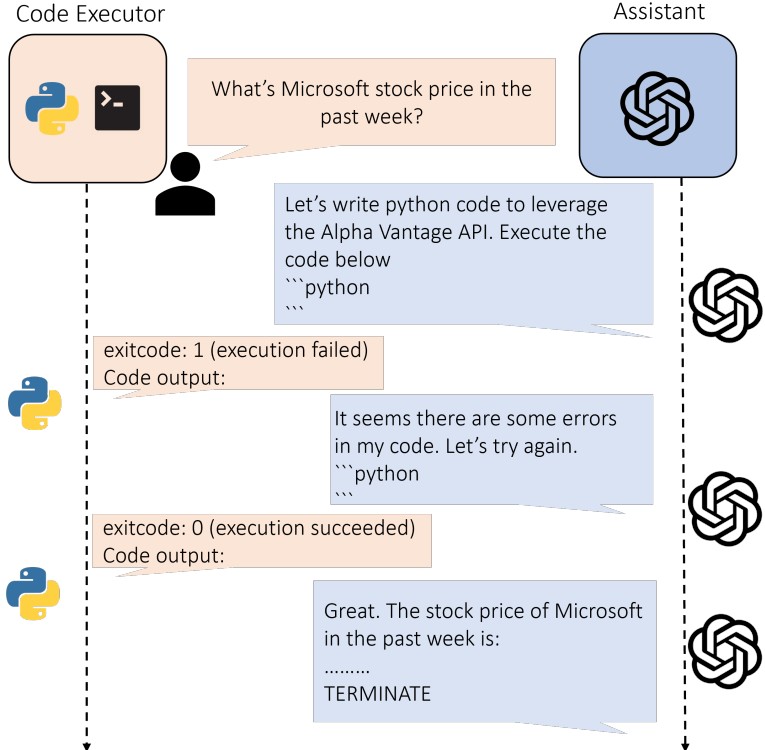

Figure 3: A visualization of the assistant-code executor conversation.

## D  IMPLEMENTATION DETAILS

**Hardware**    All experiments ran on a machine with an Intel(R) Xeon(R) CPU E5-2678 v3 with 512G memory and two 48G NVIDIA RTX A6000 GPUs. Note that the GPUs are only for hosting the LLAMA-2-13b-chat model, while the GPT family models as LLM API services do not require GPU.

**Prompts and default message**    First, we present our prompt used for each user query in Figure 4, where the red part indicates the recommended API for the LLMs to use, the blue part is the template for demonstrating the retrieved query-code pair when the solution demonstration is applied, and the final black part is the user query. For Chain-of-Thought prompting, we append one sentence "Let's think step by step." at the end of the prompt. Then, we present the system prompt we used for model evaluation in Figure 5. In particular, we leverage GPT-4 for model evaluation and set the system prompt as in Figure 5. Then we prompt the GPT-4 model with the whole conversation history, following the template stated in the first point of the system prompt. The GPT-4 model would generate a single indicator token (yes/no) as the judgment for whether the user query is successfully addressed. Finally, in the code executor agent, if there is a coding block in the response generated by the assistant, the code executor would automatically extract and execute the code, and then send back the execution results; if no coding block is detected, we set a default message "Reply TERMINATE if everything is done." as the response of the code executor to let the conversation proceed.

## E  CASE STUDY

Here, we present several case studies on how solution demonstration leads to successful query completion. In particular, we focus on the GPT-3.5-turbo assistant, and compare the conversation with and without solution demonstration. We showcase one query from each of the three datasets. The mapping between the figures of the conversation and the corresponding method/dataset is in the Table 6.

You can use the API keys in the following dictionary (key: API, value: API key):
{API_DICTIONARY}
Directly use the provided API key in your code. Do not use placeholders of API key in the code.

We provide some examples of query and python code used to solve the query below. They may be helpful as references.
----------
query: {RETRIEVED_QUERY}
code: {RETRIEVED_CODE}
----------

{USER_QUERY}

Figure 4: Prompt we used for each user query.

You are a fair AI judge.
1. You will be given a user input and a conversation history between the user and one AI assistant. Below is the schema:
  USER QUERY: <user_query>
  CONVERSATION HISTORY:
  USER 1: <user_utterance_1>
  ASSISTANT 1: <assistant_utterance_1>
  USER 2: <user_utterance_2>
  ASSISTANT 2: <assistant_utterance_2>
  ...
2. Your task is to judge whether the user's task is successfully completed.
3. The AI assistant may suggest code for the user to execute. If the user does not successfully execute the code, the task is not successfully completed.
4. Your output should be the judgement and a brief explanation of your judgement following the schema below:
  SUCCEED: Yes/No
  EXPLANATION: <explanation>

Figure 5: System prompt for model evaluation.

Table 6: The look-up table for case studies.

| Dataset | Method | Figure |
|---------|--------|--------|
| Places | GPT-3.5-turbo | Figure 6 |
|        | GPT-3.5-turbo + SolDemo | Figure 7 |
| Weather | GPT-3.5-turbo | Figure 8 |
|        | GPT-3.5-turbo + SolDemo | Figure 9& 10 |
| Stock | GPT-3.5-turbo | Figure 11 |
|        | GPT-3.5-turbo + SolDemo | Figure 12 |

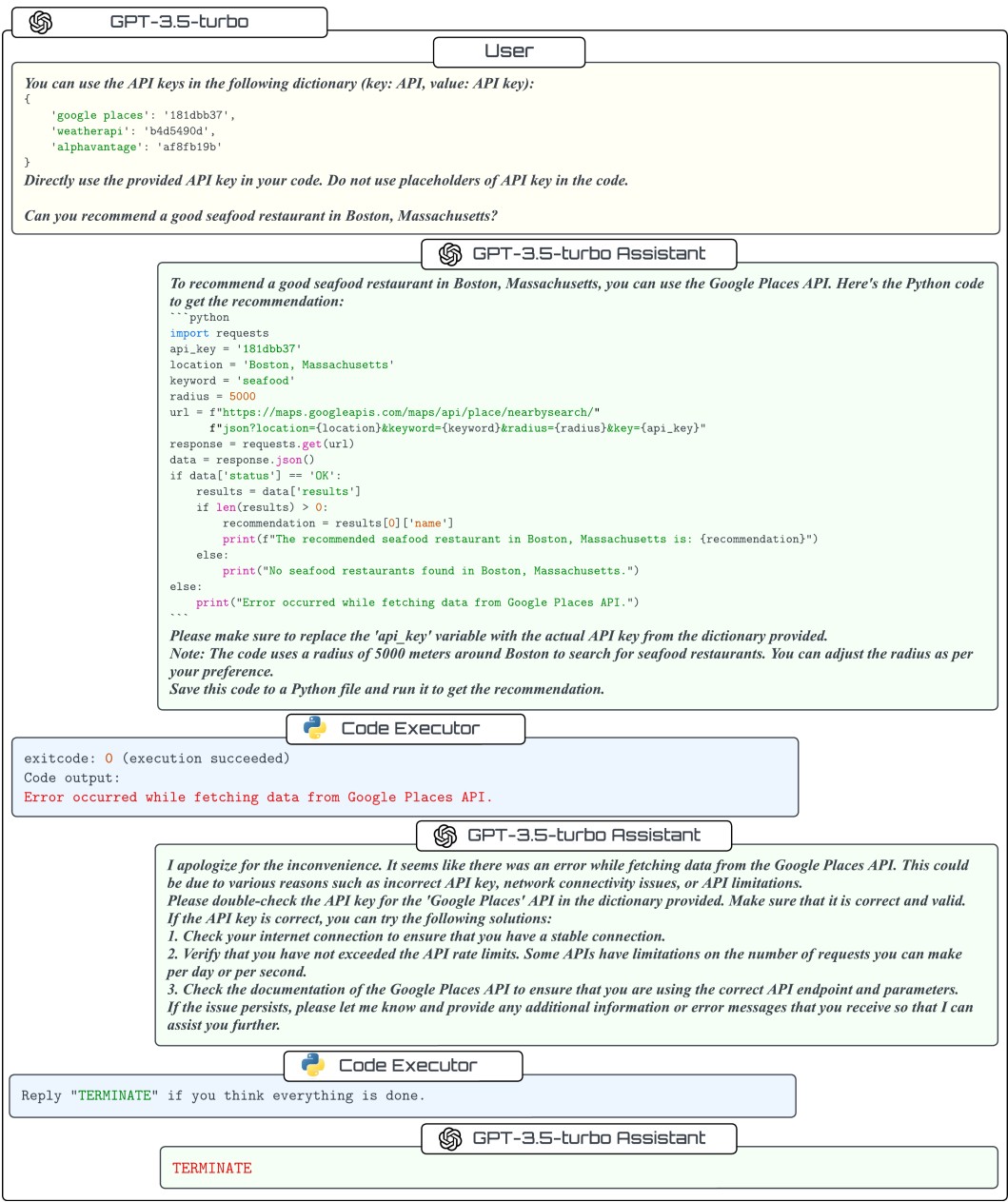

Figure 6: The GPT-3.5-turbo assistant fails to use the API due to wrong parameters.

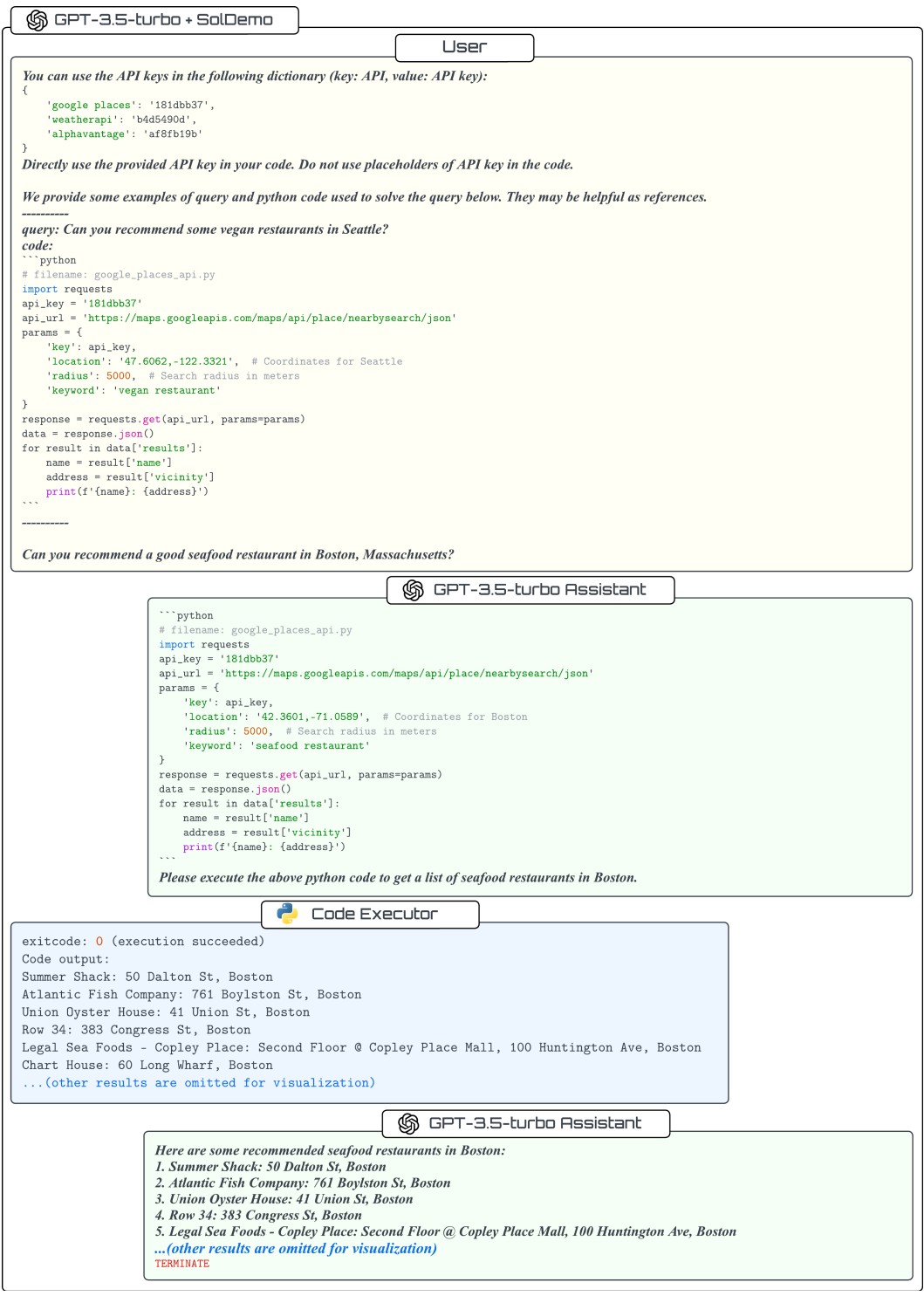

Figure 7: With solution demonstration, the GPT-3.5-turbo assistant successfully obtains the restaurant information and answers the user query.

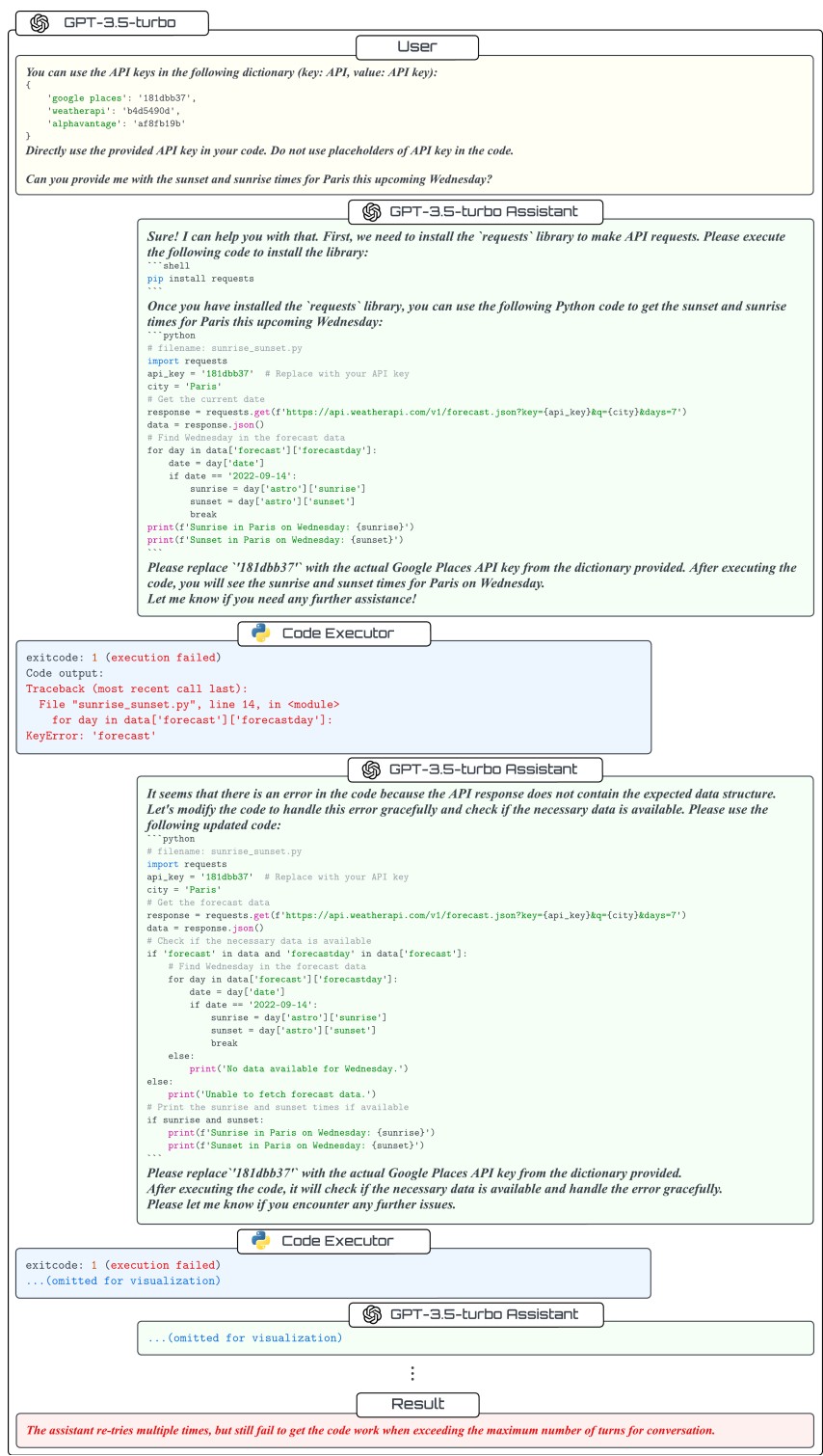

Figure 8: GPT-3.5-turbo fails to make the code work because it mistakenly use the API key of Google Places for the WeatherAPI.

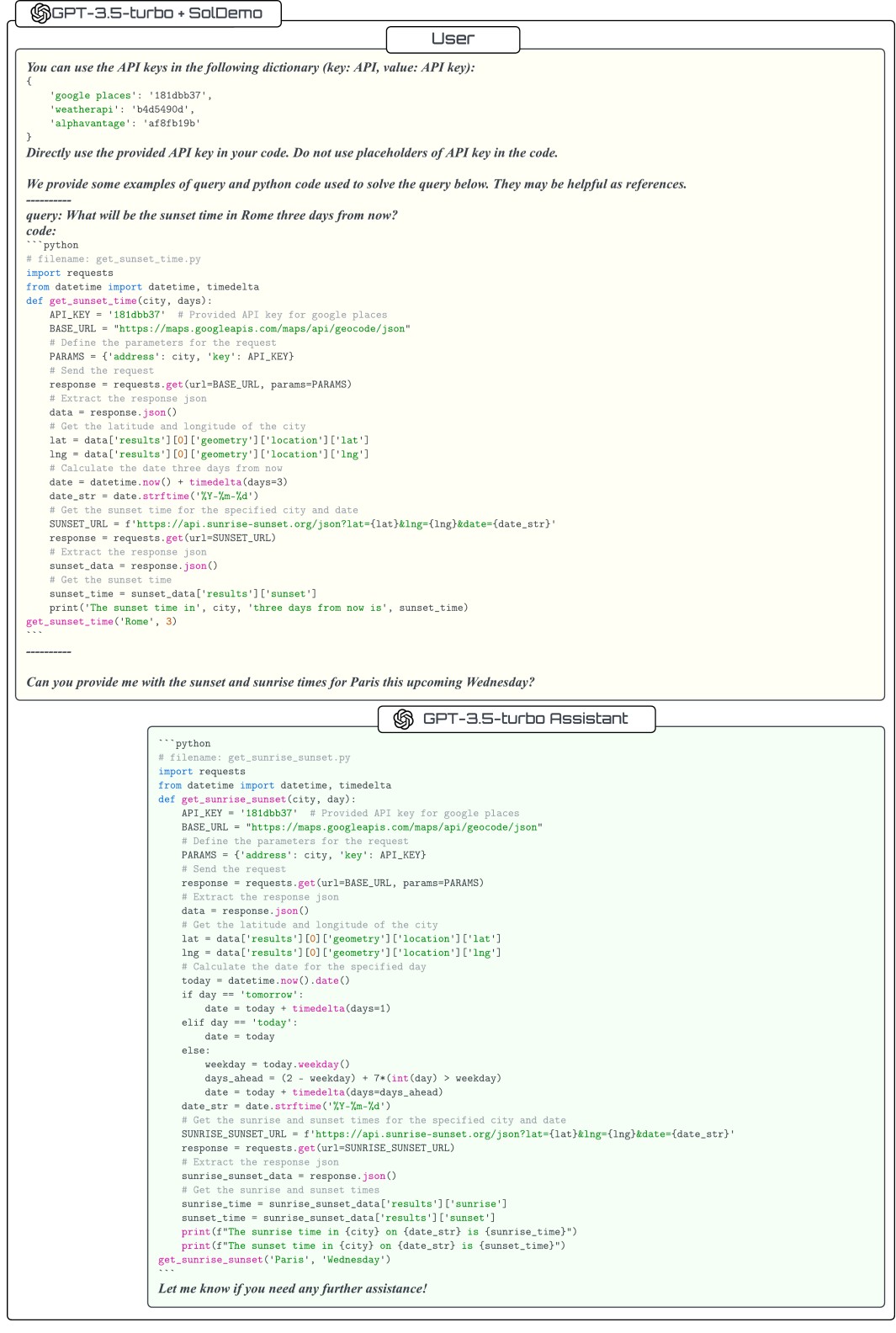

Figure 9: First half of GPT-3.5-turbo with solution demonstration for the query "Can you provide me with the subset and sunrise times for Paris this upcoming Wednesday".

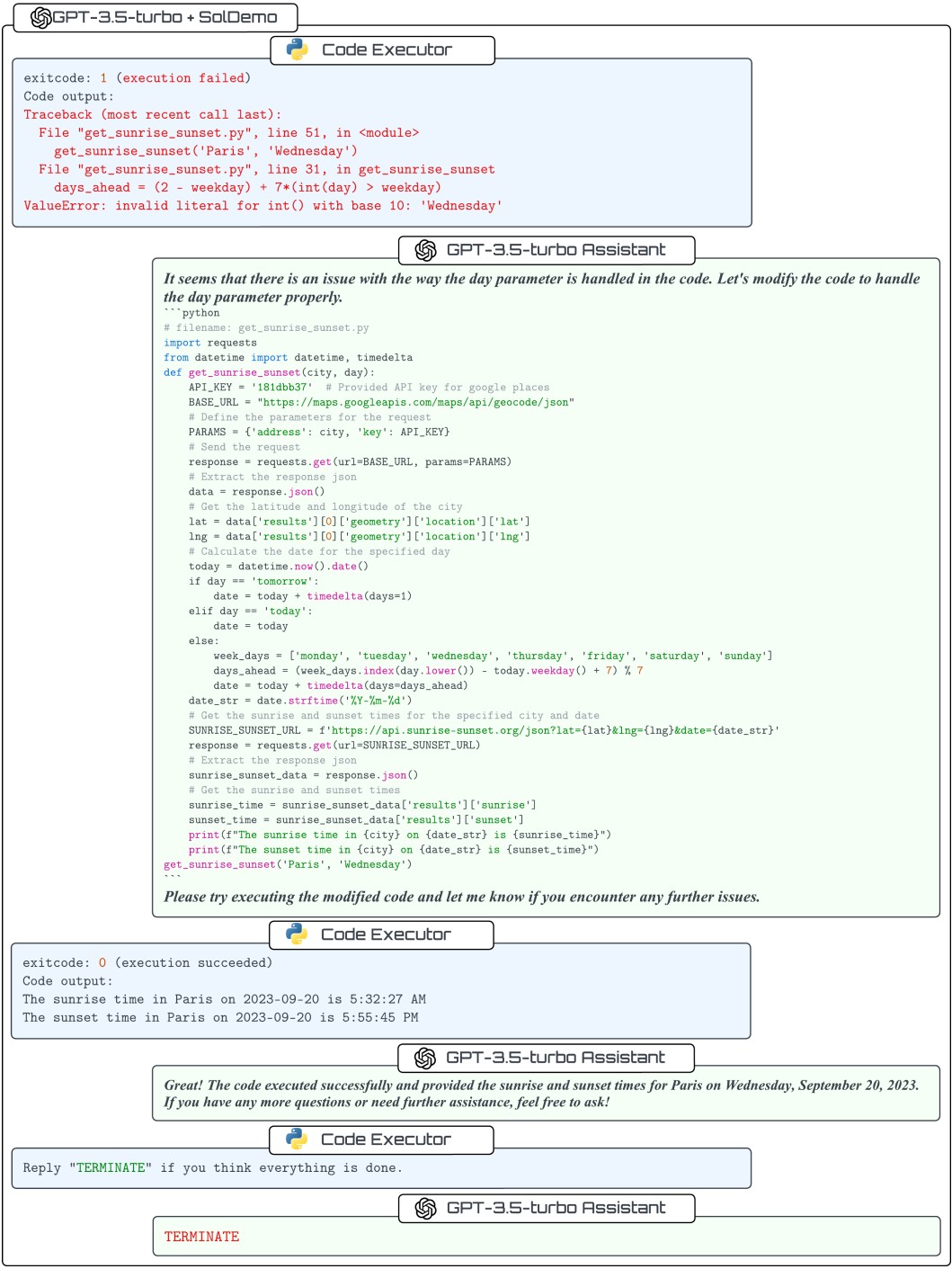

Figure 10: Second half of GPT-3.5-turbo with solution demonstration for the query "Can you provide me with the subset and sunrise times for Paris this upcoming Wednesday". With solution demonstration, GPT-3.5-turbo successfully figures out the correct date and obtains the queried information to answer the user query.

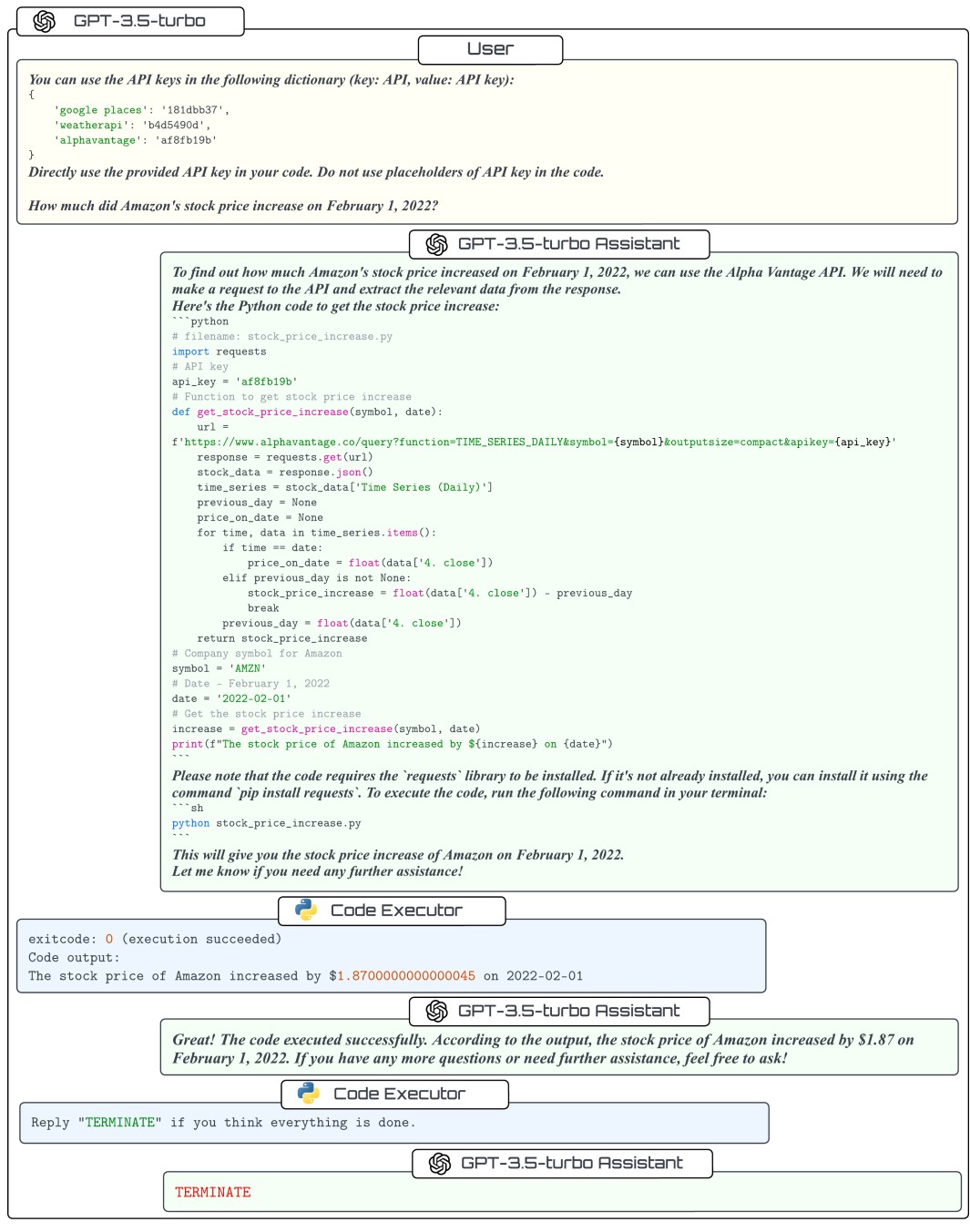

Figure 11: The GPT-3.5-turbo assistant fails to calculate the stock price rise due to incorrect code logic: the variable `price_on_date` is not used correctly.

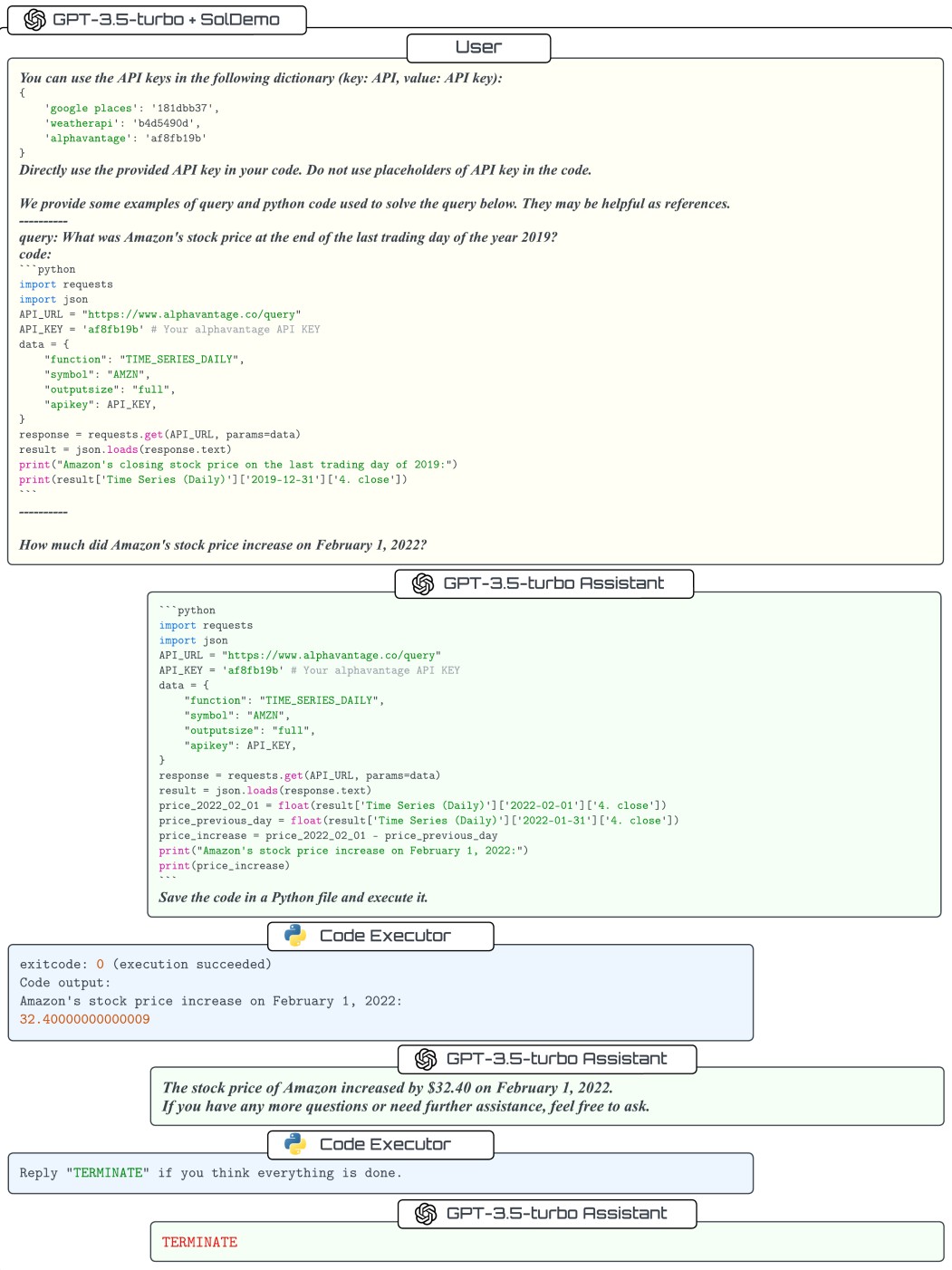

Figure 12: With solution demonstration, the GPT-3.5-turbo assistant obtains the correct stock information and answers the user query successfully.

