# OpenReview forum: "EcoAssistant: Using LLM Assistant More Affordably and Accurately"
_ICLR.cc/2024/Conference — Submitted to ICLR 2024_

### Official Review · Reviewer_UE2g · 2023-10-30

**Soundness:** 4 excellent
**Presentation:** 3 good
**Contribution:** 2 fair
**Rating:** 6
**Confidence:** 2

**Summary:**

The paper proposes an economically efficient language model agent that can interact with API. It incorporates three techniques: 1) conversationally interact with the execution environment, 2) saves cost by using a hierarchy of LLM assistant, 3) using successful demonstration as the in-context examples. Empirical results show that the proposed approach is indeed effective.

**Strengths:**

The paper proposes an economically efficient system that can interact with APIs via code. The empirical results look convincing to me.

**Weaknesses:**

This paper delivers a good system and represents a reasonable engineering contribution. However, I am a bit skeptical about its novelty: while probably no one has combined all these three tweaks together before, each of them seems relatively straightforward to me. Can fellow reviewers comment on the novelty for each of the three tweaks?

(sorry that I am not following the related works very closely so I do not know exactly how novel these ideas were; however, I think they are very straightforward ideas to try after gpt-4 release and does not require conceptual innovations)

**Questions:**

- Would you mind commenting on the novelty of the proposed approach, or say, the most surprising part of this paper? Thanks!

---

> ### Author Response · Authors · 2023-11-22
> **Thank you for your review!**
>
> Please refer to the general response for clarifications on the novelty.
>
> **Q: What is the most surprising part of this paper?**
>
> The most surprising part of this work is the effectiveness of “indirect advising” (also explained in the general response): while the assistant hierarchy is designed to reduce the cost and the solution demonstration is to improve the success rate, their combination further reduces the cost. Such a synergistic effect emerges because the solution from a more powerful model (GPT-4) would be later used to guide a weaker model (GPT-3.5) automatically without a specialized design since the query-code database is shared across assistants. By looking at the solution from the more powerful model, the weaker model could solve more tasks and therefore reduce the overall cost by reducing the reliance on the more powerful model.

---

### Official Review · Reviewer_R6M4 · 2023-11-02

**Soundness:** 3 good
**Presentation:** 3 good
**Contribution:** 2 fair
**Rating:** 5
**Confidence:** 4

**Summary:**

This paper proposes a framework, EcoAssistant for LLMs to generate API calls for answering user’s queries that require external knowledge. The framework consists of three components: iterative refinement based on automatic feedback from executors; a priority queue of LLMs where cheaper LLMs are used first; cache previously high-quality response as demonstration for further generation. The resulting system demonstrates better performance, and it’s more cost-efficient.

Overall, the design of EcoAssistant makes a lot of sense, but it lacks novelty and research depth considering many related work in this direction.

**Strengths:**

- clear presentation of the framework and results
- significant empirical improvement

**Weaknesses:**

- EcoAssistant relies on a set of known techniques (e.g., iterative refinement, demonstration library), the system per se is not novel from the technical perspective.
- from the research perspective, it does not investigate (or focus on) several key problems in this system: 1) how do you reliably collect feedback from executors? are the automatic feedbacks reliable, 2) how to decide whether a generated response is good enough to be put in the demonstration library, 3) how to design the policy for back off in the general case?

Overall, I think the design of EcoAssistant is not a significant contribution, and the authors do not go further beyond showing the empirical results of it. Though cost-efficiency is an appealing property, it’s very unclear to me what the back-off policy looks like in general.

**Questions:**

how do you decide whether a response is a success or not?  is it based on execution error?

---

> ### Author Response · Authors · 2023-11-22
> **Thank you for your review!**
>
> **Q1: how do you reliably collect feedback from executors? are the automatic feedbacks reliable?**
>
> As the executor would execute the code and send the output as feedback, the feedback is either an error message when the code has bugs or the information that the code prints. This feedback is naturally reliable because if it is an error message, it would help the assistant to debug their code like humans do; if it is printed information, it is essentially what the assistant is asking for and would help the assistant to address the user query.
>
> **Q2: how to decide whether a generated response is good enough to be put in the demonstration library**
>
> In EcoAssistant, we store a query-code pair only when the query is successfully addressed. By this way, the stored code snippets are all solutions to previous queries and therefore have good quality.
>
> **Q3: how to design the policy for back off in the general case?**
>
> Our design of the assistant hierarchy provides a general back-off policy: the order of assistant invocation is based on the cost. Specifically, it always invokes a cheaper assistant before a more expensive one. This design aims to reduce the reliance on more expensive models in order to reduce the cost of the system.

---

### Official Review · Reviewer_wYWS · 2023-11-11

**Soundness:** 2 fair
**Presentation:** 3 good
**Contribution:** 2 fair
**Rating:** 5
**Confidence:** 3

**Summary:**

In this paper, the authors present EcoAssistant, a framework for using existing LLMs to generate responses to invoke API calls in a cost effective manner and in a more autonomous manner.

**Strengths:**

- Having cost-effective solutions are useful and having this paper especially optimize for the cost is a useful strategy.

- The authors present an intuitive system that's easy to replicate, and have shown useful empirical results.

**Weaknesses:**

- I think this paper does suffer from lack of novelty. I think the paper does show an intelligent combination of existing techniques and models, but in my opinion it doesn't meet the threshold for a full paper. It would have been useful if the authors presented a methodology/algorithm that would help automatically optimize given a set of LLMs, or presented an empirical analysis on a much larger dataset with more complex APIs.

**Questions:**

- How does this method scale with # of APIs? For instance, the ToolLLM[1] paper had >16,000 APIs in their dataset. This would require some shortlisting using a retriever to make it compatible but I think adding that part would significantly help improve the novelty aspect of the paper.

- I think more error analysis would also be needed to identify what kind of queries are problematic for which models. For instance, if we can identify if smaller LLMs can easily answer easy queries then we don't need to ever invoke the larger LLMs - are you already doing this?

- Can you help me understand how do you define an exit criterion? For instance, what if the agent gets stuck in an infinite loop where the larger LLM and the smaller LLM agent keep going back and forth?


References

[1] Qin, Y., Liang, S., Ye, Y., Zhu, K., Yan, L., Lu, Y., ... & Sun, M. (2023). Toolllm: Facilitating large language models to master 16000+ real-world apis. arXiv preprint arXiv:2307.16789.

---

> ### Author Response · Authors · 2023-11-22
> **Thank you for your review!**
>
> **Q1: How does this method scale with # of APIs?**
>
> As we only provide the assistant with the API name and a fake key, we can include as many APIs as we would like as long as it can be fit into the context window. But it is possible that more APIs might confuse the model and decrease the success rate. In that case, we can directly use the API retriever from ToolLLM as a plug-and-play module for EcoAssistant. We would like to leave it to future work since our focus is the cost-effective deployment of LLM services.
>
> **Q2a: More error analysis would also be needed to identify what kind of queries are problematic for which models.**
>
> Thanks for the suggestion. We conduct error analysis on the Mixed-100 dataset (100 queries, details can be found in section 4.4), and the results are shown below. We identified 5 categories of error: 1) exceeding the context window limit of the model; 2) Inability to output correctly formatted code for execution; 3) incorrect answer or 4) partially-incorrect answer due to erroneous API use or problematic code; and 5) incorrect date (eg, asking for tomorrow’s weather but actually output today’s)
>
> | Method    | exceeds context window | code format | incorrect answer | partially-incorrect answer  | incorrect date | total number of failures
> |---------|----------|----------|---------|----------|----------|---------|
> | GPT-3.5-turbo | 3 | 56 | 14 | 1 | 1 | 75 |
> | GPT-4   | 3 | 0 | 25 | 7 | 6 | 40 |
> | EcoAssistant  | 0 | 0 | 16 | 3 | 1 | 20 |
>
> We can see that the EcoAssistant demonstrated the best performance among the three, with a total of only 20 failures. It did not exceed the context window limit or have issues with code formatting. Compared with GPT-4, it showed fewer instances of incorrect answers, partially incorrect answers, and incorrect dates. This suggests that the solution demonstration enables EcoAssistant to leverage past solutions and process queries more accurately.
>
> **Q2b: if we can identify if smaller LLMs can easily answer easy queries, then we don't need to ever invoke the larger LLMs**
>
> This idea is already realized in our assistant hierarchy: because EcoAssistant would always start with the cheapest assistant, if it succeeds, then it will not invoke more expensive assistants.
>
> **Q3: how do you define an exit criterion?**
>
> For each query, the EcoAssistant will terminate after trying out all the assistants. In particular, it always invokes the cheapest assistant first, and invokes the next one only when the current one fails to address the query. The order of invocation is based on the cost of the LLM and each assistant will be invoked at most once for each query. If the last and most expensive assistant still fails, the EcoAssistant will terminate, so it won’t get stuck in an infinite loop.

---

### Author Response · Authors · 2023-11-22
**General Response**

We thank all the reviewers for their valuable and constructive comments. As reviewers are concerned about the novelty and contribution of this work, we summarize the novelty and contributions below:

- Problem setup: 1) we focus on a new type of QA task where the model needs to write code to fetch relevant information from APIs; 2) we particularly consider the ecologically valid scenario where queries appear sequentially over time. For 1) existing work relies on pre-defined tools/functions to address queries that require external APIs, while we focus on using free-form code to fetch the information, which is more flexible and challenging, requiring no offline preparation. For 2), existing works typically consider queries as independent tasks, which is not how most real world systems will be deployed. We study how to make a system deployable by reducing its compute-cost over a sequence of streaming queries.

- EcoAssistant consists of two simple yet effective techniques: assistant hierarchy and solution demonstration. First, the assistant hierarchy technique has not been studied before; to the best of our knowledge, the most relevant work is another ICLR’2024 submission [1], where the authors present a similar idea called model cascade, but it is designed for single-turn text generation task, while we focus on the direction of LLM agents where a chat LLM has to cooperate with a code executor to address the user queries. Second, the solution demonstration is unique, as far as we know there is no existing work that presents a similar technique. The core idea is that when a query is successfully addressed, the conversation history from the two-agent system could be useful for helping solve future queries. In particular, we identify the last code snippet of the conversation history as a useful solution, and store it in a database to help with future queries.

- The two techniques collectively contribute to the “indirect advising” effect. In particular, since the query-code database is shared by the assistants, the solution from a more powerful mode (GPT-4) stored in the database would be later retrieved to guide a weaker model (GPT-3.5). Such a mechanism requires no offline preparation but could further reduce the overall cost, because by looking at the solution from the more powerful model, the weaker model could solve more tasks and reduce the invocation of more expensive models.


[1] FrugalGPT: How to Use Large Language Models While Reducing Cost and Improving Performance. ICLR submission id 6216

---

### Meta-Review · Area_Chair_xL3g · 2023-12-06

**Metareview:**

This paper proposes EcoAssistant, a system that can reduce the cost of prompting expensive LLM endpoints when answering user-issued queries using code APIs. EcoAssistant combines three ideas to reduce prompting cost. First, it uses a hierarchy of LLMs and attempts to answer a query using smaller cheaper LLMs before backing off to stronger but more expensive ones. Second, responses to past user queries from different LLMs are collectively added to a demonstration library that can be used as in-context few-shot demonstrations when solving queries that come in later. Finally, EcoAssistant uses self-refinement to iteratively improve code predictions given execution feedback. Through experiments, the authors demonstrated that EcoAssistant improves over GPT-4 in task success rate while being 50% more cost-effective.

All the reviewers concurred that the proposed three methods to reduce prompting cost are intuitive and they worked well empirically, with significant improvements over GPT-4. However, the reviewers also found a lack of novelty in this submission, as EcoAssistant “relies on a set of known techniques”, without “going further beyond the empirical results of that” (R6M4) and providing more insights into those existing approaches. While the authors argued that the idea of collectively constructing a demonstration library using an ensemble of weaker and stronger LLMs is novel during the response phase, there is not enough detailed experimental analysis in this line. Therefore, the decision is “Reject”

**Justification For Why Not Higher Score:**

This paper is a combination of existing methods, without contributing new insights out of combining those approaches. The lack of novelty and new insights makes it hard to justify accepting this paper.

**Justification For Why Not Lower Score:**

N/A

---

### Decision · Program_Chairs · 2024-01-16

Reject